# Signatures of Cholera Outbreak during the Yemeni Civil War, 2016–2019

**DOI:** 10.3390/ijerph19010378

**Published:** 2021-12-30

**Authors:** Ryan B. Simpson, Sofia Babool, Maia C. Tarnas, Paulina M. Kaminski, Meghan A. Hartwick, Elena N. Naumova

**Affiliations:** 1Division of Nutrition Epidemiology and Data Science, Friedman School of Nutrition Science and Policy, Tufts University, 150 Harrison Avenue, Boston, MA 02111, USA; paulina.kaminski13@gmail.com (P.M.K.); meghartwick@gmail.com (M.A.H.); 2Department of Neuroscience, The University of Texas at Dallas, 800 W Campbell Road, Richardson, TX 75080, USA; sfb170000@utdallas.edu; 3Department of Community Health, School of Arts and Sciences, Tufts University, 574 Boston Avenue, Medford, MA 02155, USA; maia.tarnas@gmail.com

**Keywords:** cholera, critical periods, Kolmogorov–Zurbenko filter, outbreak signature, time series, Yemen

## Abstract

The Global Task Force on Cholera Control (GTFCC) created a strategy for early outbreak detection, hotspot identification, and resource mobilization coordination in response to the Yemeni cholera epidemic. This strategy requires a systematic approach for defining and classifying outbreak signatures, or the profile of an epidemic curve and its features. We used publicly available data to quantify outbreak features of the ongoing cholera epidemic in Yemen and clustered governorates using an adaptive time series methodology. We characterized outbreak signatures and identified clusters using a weekly time series of cholera rates in 20 Yemeni governorates and nationally from 4 September 2016 through 29 December 2019 as reported by the World Health Organization (WHO). We quantified critical points and periods using Kolmogorov–Zurbenko adaptive filter methodology. We assigned governorates into six clusters sharing similar outbreak signatures, according to similarities in critical points, critical periods, and the magnitude of peak rates. We identified four national outbreak waves beginning on 12 September 2016, 6 March 2017, 28 May 2018, and 28 January 2019. Among six identified clusters, we classified a core regional hotspot in Sana’a, Sana’a City, and Al-Hudaydah—the expected origin of the national outbreak. The five additional clusters differed in Wave 2 and Wave 3 peak frequency, timing, magnitude, and geographic location. As of 29 December 2019, no governorates had returned to pre-Wave 1 levels. The detected similarity in outbreak signatures suggests potentially shared environmental and human-made drivers of infection; the heterogeneity in outbreak signatures implies the potential traveling waves outwards from the core regional hotspot that could be governed by factors that deserve further investigation.

## 1. Introduction

The ongoing cholera epidemic in Yemen is the largest in recorded history, with over 1 million suspected cholera cases since 2016 [1,2]. The inability of the Yemeni government, Houthi forces, and other involved actors and international agencies to mitigate this epidemic demonstrates the challenges that many countries face when trying to manage infectious disease outbreaks amidst other humanitarian emergencies [1,2,3,4,5]. The ongoing outbreak coincides with the Yemeni Civil War (2015–ongoing), which has crippled existing health infrastructure and depleted medical resource stockpiles [5,6]. Even before these crises, Yemen was one of the poorest countries in the Arabian Peninsula, with widespread malnutrition and rampant poverty [5].

Limited access to safe and affordable water has increased the number of individuals exposed to cholera, a waterborne disease, while a growing malnourished and immunocompromised population has increased the risk of infection [5,7]. This, coupled with limited access to health care, has severely decreased the efficacy of existing prevention and control measures. The variety of human-made and environmental drivers of the outbreak complicates what resources international aid organizations distribute and how, when, and where these resources are disseminated. The response to the ongoing cholera outbreak in Yemen differed from those in other countries, with factors governing transmission still unclear. The response is now aggravated by the COVID-19 pandemic, forcing public health professionals to rethink the ways to handle large-scale outbreaks.

The World Health Organization’s (WHO) Global Task Force on Cholera Control (GTFCC) created a global roadmap for country-level cholera mitigation in October 2017 [8,9]. The report emphasized the need for data-driven recommendations to (1) improve early outbreak detection; (2) identify regional hotspots with recurrent outbreaks; and (3) coordinate medical resource mobilization in 47 countries worldwide [8,9]. The GTFCC stressed that improved analyses of available surveillance data can improve outbreak characterization and identification of transmission patterns to create national prevention programs [8]. This included conducting routine vulnerability assessments of at-risk populations and monitoring environmental and human-made drivers of the cholera outbreak for early detection and response [8]. Once outbreak signatures are defined, public health professionals can detect common risk factors, inform the appropriate aid organizations, and improve outbreak response efforts [8,10,11].

In our early works, we explored the concept of outbreak signatures, defined as the shape of an epidemic curve in combination with its temporal features including critical points (onset, peak, resolution) and critical periods (acceleration, deceleration, steady state) [12,13,14]. We applied this concept to modeling waterborne infections and emphasized the importance of characterizing outbreak signatures for early detection and mitigation [12,13,14,15]. We also provided a methodology for estimating the peak timing and amplitude of seasonal enteric infections and used these features to assess the synchronization of infections across geographic regions [16,17,18,19,20,21,22,23,24].

While researchers agree on a general definition of an outbreak and its features, such as onset, peak intensity, and resolution, there are no widely accepted systematic approaches to estimate such features from historic data. The onset and resolution timing are often declared (and occasionally revised) by local, national, or international officials. The common approach to defining the outbreak signature is solely based on the visual inspection of an epidemic curve or basic agreements [25]. We expanded on our early works by proposing a methodology for systematically defining outbreak signatures and classifying regional hotspots to explore the potential of traveling waves of infection outwards from regional hotspots.

In this study, we quantified outbreak features, including critical points (onset, peak, and resolution) and critical periods (acceleration, deceleration, and steady state), of the ongoing cholera epidemic in Yemen and clustered governorates according to these features. We compiled publicly available data from the WHO’s Eastern Mediterranean Regional Office’s (EMRO) epidemiological bulletins and created a weekly time series of cholera rates for 20 of 21 governorates in the period 2016–2019. We defined critical points and periods using a non-parametric Kolmogorov–Zurbenko adaptive filter and classified governorates according to similarities of these features. Our curated time series dataset and applied methodology could offer technical guidance to supplement the GTFCC cholera mitigation roadmap and can be used to further investigate human-made and environmental drivers of cholera outbreaks in conflict areas.

## 2. Materials and Methods

### 2.1. Data Sources

The WHO, in collaboration with the Yemeni Ministry of Health (MOH), has collected records of laboratory-confirmed cholera infections in Yemen as part of the electronic Disease Early Warning System (eDEWS) since 2013 [26]. This database contains public and private health facility records compiled into a district-level time series dataset based on the time of hospital admission [26,27]. The MOH and the WHO reported governorate-level aggregated eDEWS data via the WHO’s Eastern Mediterranean Regional Office’s (EMRO) epidemiological bulletins for public use in the period 2016–2019 [28,29,30].

These bulletins changed reporting formats multiple times during the study period, specifically its temporal resolution (as daily, weekly, and monthly cumulative counts) and spatial granularity (at the governorate and national levels). No epidemiological bulletins were available prior to Week 36 of 2016. We had only 9 weeks (5% of 172) with missing data. Records were missing for 4 consecutive weeks in 2017 (Week 13–16 (27-March to 23-April)), 1 week in February of 2018 (Week 8 (19–25 February)) and 4 consecutive weeks in July of 2018 (Week 27–30 of 2018 (2–29 July)). We excluded Hadramaut and Socotra island from our analysis as the EMRO inconsistently reported data for these locations. The EMRO did not report cholera data for 2020 or 2021. All discovered bulletins along with the raw and curated records were uploaded to figshare [31].

### 2.2. Time Series Compilation

To conduct our analyses, we compiled all abstracted records in a uniform format. We aggregated data by WHO-defined epidemiological weeks as performed for other WHO-monitored infections [32]. Daily case reports were available for 29 weeks (17% of study period) from Week 36 of 2016 (4 September 2016) to Week 12 of 2017 (21 March 2017). Weekly reports were available for 62 weeks (36%) from Week 17 of 2017 (24 April 2017) through Week 26 of 2018 (1 July 2018). Monthly reports were available for 74 weeks (43%) from July 2018 through December 2019. We describe the procedure for time series compilation below.

We extracted daily case data from irregularly reported WHO EMRO daily epidemiological bulletins and aggregated cases by WHO-defined weeks. Bulletins provided updates on the number of cumulative confirmed infections occurring during multi-day reporting periods based on time of hospital admission [26,27]. We estimated average daily cases by subtracting total cases from consecutive reports and dividing by the number of days in the reporting period. We prorated weekly case counts for Week 12 of 2017 as records were incomplete (5 of 7 days had missing records). We interpolated records for 4 consecutive weeks (14% out of 29) using linear approximation based on information from adjacent weeks (before and after the missing week) capturing the overall seasonal behavior.

We extracted weekly records from epidemiological bulletins available from Week 17 of 2017 (24 April 2017) through Week 26 of 2018 (1 July 2018). Weekly data were reported using a governorate-level spreadsheet containing cumulative confirmed cholera cases for the WHO-defined “Second Wave” of the epidemic. In these bulletins, weekly cases were provided for the week of the report, including both provisional and confirmed cases, and for each of the prior three weeks, which included only confirmed cases. We reviewed each weekly bulletin in chronological order and extracted the 3 week lagged confirmed case estimate that presumably contained most accurate information. For 5 incomplete bulletins (8% out of 62), we estimated weekly cases using 2 week and 1 week lagged case estimates. We interpolated cases for Week 8 of 2018 (19–25 February) using adjacent weeks’ values.

We used monthly records from the WHO EMRO monthly situation reports available from July 2018 through December 2019. These bulletins provided cumulative confirmed cases from Week 17 of 2017 to the final day of the reporting month. We converted monthly cumulative counts to weekly case counts in two steps. First, we subtracted consecutive monthly bulletins to estimate newly reported cases in that month. Next, we estimated average cases for each day within a given month by dividing these new cases by the number of days within the month, and aggregated cases according to WHO-defined epidemiological weeks. We interpolated missing records for 4 consecutive weeks (5.6% out of 72) using linear approximation based on information from adjacent weeks.

We estimated national cases by summing all governorate-level cases for each week of the 172 week study. We conducted all analyses according to study week (ranging 1–172) and reported all results using a combination of WHO-defined epidemiological weeks (ranging 1–52 per year) and Gregorian calendar dates (formatted DD-Month-YYYY).

### 2.3. Rate Calculations

To compare outbreak signatures across governorates, we calculated weekly rates of infection using fatality-adjusted, pro-rated weekly population estimates. We extracted data on conflict fatalities using the Armed Conflict Location and Event Data (ACLED) project, which reported on both violent and non-violent political events [33,34]. We summed all fatalities from all political events on each day per governorate and then aggregated daily totals by WHO-defined epidemiological week. We used fatality counts to correct the population estimates. We also applied a low-to-moderate population growth rate (≈ 0.024) reported by the Yemeni Central Bureau of Statistics to approximate population changes during the civil war [35] and prorated this growth rate for the annual cycle in weeks (e.g., 0.024/52.25 = 0.00045933).

We estimated weekly governorate-level population in several steps. First, we calculated the population for Week 1 of 2017 as the average of population estimates from multiple sources including the WHO EMRO, the 2004 Yemeni Central Statistical Organization (CSO) population projection for 2005–2025, and the International Organization for Migration (IOM) Displacement Tracking Matrix’s (DTM) 2017 Population Estimate [30,35,36]. Next, for Week 2 of 2017 through Week 52 of 2019, we estimated weekly governorate-level population iteratively forwards by subtracting conflict fatalities from the prior week and multiplying the difference by the prorated annual growth rate (Equation (1)). Finally, to correct for fatalities occurring between Week 36 of 2016 and Week 1 of 2017, we estimated weekly population by adding conflict fatalities and dividing the sum by the weekly-adjusted population growth rate (Equation (2)).
(1)$$P_{s,t}=\left(P_{s,t-1}-F_{s,t-1}\right)*\left(1+\frac{0.024}{52.25}\right)\;\;\mathrm{if}\;t\;>\;\mathrm{Week}\;1\;\mathrm{of}\;2017$$
(2)$$P_{s,t}=\left(P_{s,t+1}+F_{s,t+1}\right)*\left(1-\frac{0.024}{52.25}\right)\;\;\mathrm{if}\;t\;<\;\mathrm{Week}\;1\;\mathrm{of}\;2017$$
where *P_s,t_*—the population for *s*-governorate in *t*-week; and *F_s,t_*—the all-cause war conflict fatalities for *s*-governorate in *t*-week.

We calculated weekly rates of confirmed cholera infections (cases per 100,000 persons or cph) by dividing weekly cases by population estimates and multiplying by 100,000. We provide the curated weekly time series of case and rate estimates with and without interpolated values (Appendix A).

### 2.4. Defining Outbreak Signatures and Features

We defined outbreak signatures using the timing of critical points, the duration of critical periods, and the magnitude of smoothed peak rates (example shown in Figure 1; Appendix A). Critical points included the onset (O) of an outbreak when weekly rates began increasing, an outbreak peak (P) when weekly rates reached local maxima, and the return-from-peak or resolution (R) of an outbreak when weekly rates returned to steady-state levels, if this occurred.

Critical periods included the acceleration period from onset to peak, the deceleration period from peak to resolution (or next outbreak onset if no resolution occurred), and the steady-state period from resolution to onset. Each period had a distinct behavior: acceleration periods indicated times when weekly rates steadily increased, deceleration periods indicated when rates steadily decreased, and steady-state periods indicated when rates plateaued. We also calculated the pace of increase (proxy to basic reproduction number) and pace of decline as the linear approximation from onset timing to peak timing and peak timing to resolution timing or next wave onset timing, respectively. We reported critical period duration in weeks and pace of increase or decline in cph per week (cph/week). By comparing the timing of critical points, duration of critical periods, and magnitude of smoothed peak rate times in different locations, we defined regional hotspots and explored the potential of traveling waves of infection outwards from regional hotspots.

To produce smoothed outbreak signatures and detect change points in the time series (including the time periods when the absolute change increased, decreased, or approached near-zero values), we applied a modified version of the Kolmogorov–Zurbenko (KZ) adaptive filter [37,38,39,40]. The KZ filter is a non-parametric smoothing technique used to detect distinct changes in a highly noisy time series [37,38,39,40]. This technique used a real-values time series X(t), t=0,±1,±2,±3… to generate a set of moving averages (MA) for windows of various sizes (Equation (3)):(3)$$Z\left(t,q\right)=\frac{1}{2q+1}\;\overset{q}{\underset{l=-q}{\sum }}X\left(t,l\right)$$
where *q* was the half length of a MA window, so Z(t,q) were the smoothed values of X(t) with a different degree of reduced noise.

We created a set of time series of natural logarithm (ln)-transformed corrected rates (Equation (4)), estimated as:(4)$$Y_{t,s}^{*}=\ln\left(\frac{C_{t,s}+\alpha }{P_{t,s}}*M\right)$$
where *C_t,s_* were weekly counts for *t*-week and *s*-location corrected by *a* = 1 case and adjusted for estimated weekly population (*P_t,s_*) estimates, per *M* = 100,000 persons. Natural logarithm transformations addressed the right skewed distribution of weekly rates due to short bursts of high values during epidemic peaks and prolonged periods of low reported rates in the beginning of the epidemic. We then created a set of time series of the trivial derivatives, which were estimated as the absolute change in weekly ln-transformed rates: $Y_{t,s}^{*\prime }=Y_{t+1,s}^{*}-Y_{t,s}^{*}$.

We applied the KZ filter iteratively and generated a set of MA for both sets (Equations (5) and (6)), as:(5)$$Z_{t,s,q}^{*}=\frac{1}{2q+1}\;\overset{q}{\underset{l=-q}{\sum }}Y_{t,s,l}^{*}$$
(6)$$Z_{t,s,q}^{*\prime }=\frac{1}{2q+1}\;\overset{q}{\underset{l=-q}{\sum }}Y_{t,s,l}^{*\prime }$$
for *q* = 1–25, to cover various window sizes ω=2q+1 , from 3 to 51 weeks to depict the outbreak signature and to reflect the temporal changes from about one month to about one year. We then examined each set of select windows that sufficiently reduced noise yet maintained inflection points within the trivial derivative, as a verification step. By comparing the fit with sum of squares criteria, we selected candidates averaging across four windows: *q* = 1–15, *q* = 1–11, *q* = 1–7, and *q* = 3–5. We selected the best performing smoother (*q* = 3–5) to assess the similarities across governorate-specific outbreak signatures and to estimate the absolute change in weekly smoothed rates, as: $Z_{t,s,3\text{–}5}^{\prime }=\left(Z_{t+1,s,3}^{*}+Z_{t+1,s,5}^{*}\right)/2-\left(Z_{t,s,3}^{*}+Z_{t,s,5}^{*}\right)/2$. The fit of this smoother for each governorate and nationally is shown in Appendix A.

We estimated local maximums and minimums in $Y_{t,s}^{*},\;\;Z_{t,s,3\text{–}5}^{*},\;Z_{t,s,0\text{–}7}^{*},\;\;Z_{t,s,0\text{–}11}^{*},\;Z_{t,s,0\text{–}15}^{*},$ to identify time points and periods of high and low disease rates, respectively. To confirm maximums and minimums detected in the smoothed values of rates, we identified weeks when $Y_{t,s}^{*\prime },\;\;Z_{t,s,3\text{–}5}^{*\prime },\;Z_{t,s,0\text{–}7}^{*\prime },\;\;Z_{t,s,0\text{–}11}^{*\prime },\;\mathrm{and}\;Z_{t,s,0\text{–}15}^{*\prime }$ approached zero. We defined outbreak onset as the beginning of a rapid rise of rates ($Z_{t,s,3\text{–}5}^{*}$) and inflection of the trivial derivative ($Z_{t,s,3\text{–}5}^{\prime }$) from near-zero to high positive values. We defined outbreak peak as the time when rates were reaching a local maximum of $Z_{t,s,3\text{–}5}^{*}$ with the inflection point of $Z_{t,s,3\text{–}5}^{\prime }$ at near-zero or negative values. We defined outbreak resolution time as when rates $Z_{t,s,3\text{–}5}^{*}$ were low or near a local minimum and $Z_{t,s,3\text{–}5}^{\prime }$ approached near-zero values. We expressed critical time points and periods in WHO-defined epidemiological weeks along with its plausible range.

We defined outbreak waves as either the duration from onset to resolution critical points or the duration between two onset critical points if no definite resolution occurred using the most refined window of 3–5 weeks. If no resolution occurred, we differentiated between waves by identifying weeks when rates $Z_{t,s,3\text{–}5}^{*}$ plateau for 4–6 weeks followed by 3–5 weeks of a steady increase in rates when the trivial derivative ($Z_{t,s,3\text{–}5}^{\prime }$) increased from near-zero to high positive values. When possible, we identified primary and secondary outbreak peaks for specific waves based on our ability to detect local minimums and maximums. Secondary peaks referred to times when rates reached a local maximum of $Z_{t,s,3\text{–}5}^{*}$ after a burst of $Z_{t,s,3\text{–}5}^{\prime }$ from high negative to high positive values. To confirm the potential for a secondary peak, we calculated the acceleration period and pace of increase from wave onset to each peak timing and calculated the deceleration period and pace of decline from each peak timing to the conclusion of the wave (resolution or following wave onset) that are characteristic for an ongoing outbreak.

### 2.5. Clustering of Outbreak Signatures

We applied pairwise Spearman cross-correlation estimates, covering six lead and six lag weeks (or lead-lag of +6 to −6 weeks) to examine similarities across governorate-specific outbreak signatures using the best performing smoother (Appendix A). We selected this lead-lag structure because governorate critical points varied by approximately six weeks. We assigned governorates with strong serial cross-correlations, close spatial proximity to each other, and similarities across outbreak signature features including the timing of critical points, duration of critical periods, and magnitude of peak rates into the same cluster (six clusters in all).

We performed all statistical analyses and visualizations using Excel (14.3.6), Stata (SE 15.1), and R (3.6.3) software. We created all maps using publicly available shapefiles reported by the United Nations (UN) Office for the Coordination of Humanitarian Affairs (OCHA) and published on the Humanitarian Data Exchange (HumData) [41]. The data and R codes used to create maps and illustrations in this study are available in Appendix A.

## 3. Results

We presented results starting with quantifying outbreak features for Yemen nationally and by governorate using smoothed rates and trivial derivatives. We then examined similarities in outbreak features (critical points, critical periods, spatial proximity, and smoothed rate magnitudes) to define clusters. Finally, we analyzed similarities within and differences between clusters on a one-by-one basis in all subsequent subsections.

### 3.1. Four Waves of Cholera Outbreak Signatures

Over our 172 week study period, we identified ~2.1 million confirmed cholera infections among the ~28.6 million estimated inhabitants of Yemen. Based on the estimated onset, peak, and resolution critical points and acceleration, deceleration, and steady-state period durations for the Yemeni cholera outbreak from 4 September 2016 through 29 December 2019, we identified four distinct national outbreak waves (Figure 2, Table 1 and Table 2; Appendix A). Wave 1 began in 12–18 September 2016 with a peak in 7–13 November 2016 and resolution in 2–8 January 2017. Peak rates were low during this wave (peak rate ~0.06 cph) and the outbreak only affected 15 governorates. Wave 2, the largest outbreak wave nationwide, began approximately nine weeks after the conclusion of Wave 1 (6–12 March-2017). We identified two peaks in Wave 2: the first in 24–30 July 2017 (peak rate ~151.3 cph) and the second in 18–24 September 2017 (peak rate ~141.1), with a sharp decline in rates between peaks (~20% of peak rate values). The speed of increase for the primary and secondary peaks was 1.88–2.32 cph/week and 1.05–1.76 cph/week greater than the speed of decline, respectively. The prolonged decline from the second peak to the time of resolution (2–8 April 2018) lasted 28 weeks. For both Waves 1 and 2, the acceleration period from the onset to the peak was 1.5–2.2-fold shorter than the deceleration period from the peak to the onset of the next wave (Table 2).

We also identified two additional waves: Wave 3 from 28 May 2018 through 27 January 2019 and Wave 4 from 28 January 2019 through 29 December 2019. In Wave 3, we identified two peaks: the first in 22–28 October 2018 (peak rate ~53.1 cph) and the second in 17–23 December 2018 (peak rate ~45.6 cph). These peaks were approximately 66% lower than the peak rate values for Wave 2, declined briefly (~10 cph) in rate values between peaks, and had a speed of increase/decline of ~2 cph/week. For Wave 3, the acceleration period from the onset to the peak was >1.5-fold longer than the deceleration period from the peak of Wave 3 to the onset of Wave 4. For Wave 4, we identified three peaks (22–28 April 2019, 15–21 July 2019, 16–22 September 2019). These peaks declined in magnitude throughout the outbreak wave (~99.7 cph, ~73.4 cph, ~66.1 cph) and occurred approximately 12 weeks apart. We did not identify a resolution critical point for Wave 4 by the end of our study.

### 3.2. Governorate-Level Signature Variability

In our study period, Al-Hudaydah governate had the most reported cases cumulatively (n = 318,969) followed by Sana’a City (n = 230,577), Dhamar (n = 205,179), and Sana’a (200,003). Both Al-Hudaydah and Sana’a City were among the most populated governorates during this period (average weekly population of ~3.3 million and ~3.0 million persons, respectively) as well as Taizz (~3.1 million persons) and Ibb (~3.0 million persons). In the center and bottom panels of Figure 2, we provide a heatmap of smoothed rate values and their trivial derivative estimates for each governorate, respectively. Both heatmaps show that outbreak signatures varied substantially across governorates. The Wave 1 onset timing spanned from 12 to 18 September 2016 in Sana’a to 14–20 November 2016 in Al-Jawf with peak rates <1 cph in all governorates. Yet, in all 15 affected governorates, Wave 1 was resolved by 2–8 January 2017.

We identified a uniform Wave 2 onset in 6–19 March 2017 for all 20 governorates. Eight governorates had one outbreak peak during Wave 2, which occurred from 17 July 2017 to 8 October 2017. Nine governorates had two outbreak peaks with the secondary peak occurring either from 18 September 2017 to 31 December 2017 or from 22 January 2018 to 25 March 2018. Three governorates (Dhamar, Al-Bayda, and Sana’a) had three outbreak peaks. We also found variation in the number and timing of peaks for Wave 3 (12 governorates with one peak, 6 governorates with two peaks) and Wave 4 (1 governorate with one peak, 8 governorates with two peaks, and 11 governorates with three peaks).

We found groups of governorates that were strongly correlated with one another according to Spearman correlation estimates. Sana’a and Sana’a City were strongly correlated with Al-Hudaydah (ρ = 0.82–0.90; *p* < 0.001) for a lag of +1–5 weeks and with each other (ρ = 0.86–0.88; *p* < 0.001) for a lead-lag of −3 to +1 weeks. We found even stronger correlations between Amran, Al-Mahwit, and Dhamar (ρ = 0.87–0.96; *p* < 0.001) for a lead-lag of −2 to +4 weeks. We also found strong correlations between Raymah and Ibb (ρ = 0.82–0.84; *p* < 0.001), Raymah and Taizz (ρ = 0.86–0.88; *p* < 0.001), Ibb and Taizz (ρ = 0.88–0.94; *p* < 0.001) for a lead-lag of −2 to +2 weeks.

### 3.3. Six Clusters of Outbreak Signatures

Based on the combination of correlation estimates, timing of critical points, and geographic proximity, we assigned all governorates into one of six clusters including a core outbreak cluster, two immediately neighboring clusters, and three remote clusters in northern, southern and eastern regions of Yemen (Figure 3; Appendix A).

### 3.4. Core Cluster

We identified Sana’a, Sana’a City, and Al-Hudaydah governorates as the core outbreak cluster (Figure 4; Appendix A). We used the term ‘core’ as these governorates were located centrally in Yemen, contained or surrounded the nation’s capital, housed a large percentage of the population, and had the earliest timing of onset and peak critical points in Wave 1. These governorates had closely aligned primary and secondary peaks in Wave 2 from 26 June 2017 to 30 July 2017 (131.4–172.8 cph) and from 18 September 2017 to 1 October 2017 (82.1–215.8 cph). Pace of increase for primary peaks were 2.43–2.75-fold greater than pace of decline for all governorates. After a long deceleration period of ~30 weeks, cholera rates never returned to pre-Wave 2 baseline levels for any of these three governorates. Rates decreased by ~40–66% for these governorates during the ~6 weeks between peaks. While Sana’a had a smaller third peak (66.8 cph) in 1–7 January 2018, Sana’a City and Al-Hudaydah had a brief (9–14 weeks) resolution period from 19 March 2018 to 1 July 2018.

In Wave 3, all three governorates had two peaks: from 24 September 2018 to 21 October 2018 and from 19 November 2018 to 30 December 2018. For each governorate, primary and secondary peak rate values were nearly identical (Sana’a: ~112 cph; Sana’a City: ~51 cph; Al-Hudaydah: ~81 cph). In Wave 4, these governorates had the same first peak timing in 15–28 April 2019. However, the second peak in Sana’a and Sana’a City occurred in 22–28 July 2019, approximately two months prior to the third peak of Sana’a City and second peak of Al-Hudaydah in 16–22 September 2019. Differences in the alignment of peak timing supported Sana’a and Sana’a City’s strong association (ρ = 0.82–0.89; *p* < 0.001) with Al-Hudaydah for lags from 1 to 5 weeks.

### 3.5. Immediate Neighboring Cluster I

We identified four governorates, Amran, Al-Mahwit, Dhamar, and Al-Bayda, that formed a cluster immediately neighboring the core cluster (Figure 5; Appendix A). This neighboring cluster had similar critical points but higher peak rates in Wave 2 compared to the core cluster.

Though the early primary peak (24 July 2017 to 27 August 2017) was similar to the core cluster, the neighboring cluster had a later secondary peak from 5 March 2018 to 1 April 2018 that fell ~10 weeks after the core cluster. Additionally, Al-Mahwit, Al-Bayda, and Amran had peak rates of ~400 cph, roughly twice the magnitude of peak rates in the core cluster. The pace of increase for primary peaks ranged from 10.8 to 20.9 cph/week, which was on average twice as large compared to the core cluster (6.6–10.8 cph/week). Unlike the core cluster, none of the governorates in this neighboring cluster had a Wave 2 resolution, though rate values declined to ~20 cph (roughly twice the magnitude of the core cluster) by the onset timing of Wave 3.

In Wave 3, Amran, Al-Mahwit, and Dhamar had only one peak, which occurred from 24 September 2018 to 21 October 2018. Al-Bayda had a secondary peak in 19–25 November 2018, which was identical in timing to governorates in the core cluster. As in Wave 2, Wave 3 peak rate values for the neighboring cluster were, on average, higher than the peak rates for core cluster governorates (63.1–188.4 cph).

In Wave 4, all governorates peaked twice (except Al-Bayda, which peaked three times). All four governorates shared a secondary peak in 16–29 September 2019, which aligns with the later peak timing for Sana’a City and Al-Hudaydah.

### 3.6. Immediate Neighboring Cluster II

Raymah, Ibb, and Taizz formed a second cluster that immediately neighbored the core cluster. This second neighboring cluster had two well-aligned peaks in Wave 2, one well-aligned peak in Wave 3, three peaks in Wave 4, and low peak rate magnitudes compared to previously described clusters (Figure 6; Appendix A).

In Wave 2, the primary peak occurred from 26 June 2017 to 23 July 2017 (like the core cluster) and the secondary peak occurred from 15 January 2018 to 11 March 2018 (like the first neighboring cluster). Wave 2 peak rates ranged from 9.25 to 125.6 cph, which were approximately half the magnitude of the first neighboring cluster (216.3–397.7 cph). All governorates in this second neighboring cluster reached resolution timing for Wave 2 in 2–15 April 2018, when rates nearly returned to pre-Wave 2 levels (~2.45–3.56 cph).

In Wave 3, the governorates in this neighboring cluster had only one outbreak peak, which occurred from 24 September 2018 to 25 November 2018 (same as core cluster). However, peak rates had low magnitudes, ranging from 22.3 to 47.2 cph, which were less than half the peak rate magnitudes for governorates in the core and first neighboring clusters. In Wave 4, this neighboring cluster had similar peak timing as the core cluster for all three identified peaks.

### 3.7. Remote Clusters: Northern, Southern, and Eastern

The ten remaining governorates had outbreak signatures that differed from the governorates in the core and neighboring clusters, forming three geographically distinct clusters in the northern (Hajjah, Al-Jawf, and Sa’ada), southern (Al-Dhale’e, Abyan, Aden, and Lahj), and eastern (Marib, Al-Maharah, and Shabwah) regions of Yemen. Governorates within these remote clusters had one peak in Waves 2 and 3, a short Wave 2 deceleration period duration, and an early arrival of Wave 2 resolution critical points compared to the other three clusters.

In the northern cluster (Figure 7; Appendix A), governorates had Wave 2 peak timing from 4 September 2017 to 8 October 2017. Rate values reached magnitudes like those in the core outbreak cluster (73.0–208.1 cph). However, the pace of increase to and pace of decline from Wave 2 peak timing were approximately the same and occurred ~6–10 weeks after the primary peak timing of governorates in the first neighboring cluster. Cholera rates declined to pre-Wave 2 magnitudes (<1 cph) 16–30 weeks after the primary peak. Governorates within this northern cluster had Wave 3 peak rate magnitudes (24.0–49.0 cph) like those in the second neighboring cluster governorates.

In the southern cluster (Figure 8; Appendix A), Al-Dhale’e, Abyan, and Aden had only one Wave 2 peak that occurred in 17–23 July 2017. Lahj had two Wave 2 peaks: the primary occurred in 14–20 August 2017 and the secondary occurred in 25–31 December 2017. Irrespective of peak timing, all governorates had a prolonged deceleration period of ~30 weeks and reached Wave 2 resolution from 22 January 2018 to 18 March 2018. By the conclusion of Wave 2, governorates within this cluster had returned to pre-Wave 2 rate magnitudes (<1 cph).

Governorates in the eastern cluster (Figure 9; Appendix A) had no reported cases of cholera during Wave 1 and no well-defined critical points in Wave 3. In fact, cholera rates did not exceed 5 cph between the resolution of Wave 2 (as early as 18–24 December 2017) and the onset of Wave 4 (as late as 18–24 March 2019) for any eastern cluster governorate. In Wave 2, these governorates had a wide range of peak rate magnitudes, from Shabwah at ~22.1 cph to Marib at ~110.2 cph. Shabwah and Al-Maharah had the earliest resolution timing, which occurred from 18 December 2017 to 4 February 2018. As with governorates in the southern cluster, all governorates in the eastern cluster had Wave 4 peak rate magnitudes of <35 cph.

## 4. Discussion

Our study demonstrated the utility of publicly reported surveillance data to characterize, classify, and compare infectious disease outbreak signatures. We proposed an application of non-parametric KZ adaptive filters to define outbreak signatures and identified regional hotspots of cholera outbreaks. We used this data-driven approach to define outbreak critical points (onset, peak timing, and resolution) and critical periods (acceleration, deceleration, and steady state) to examine spatiotemporal patterns in cholera outbreak signatures. This methodology introduced a protocol for routine vulnerability mapping of outbreak hotspots to improve resource management and mobilization during humanitarian aid responses.

Application of KZ filter methodology permitted closer inspection of subtle differences between governorates’ outbreak signatures considering several investigated features. Using the proposed methodology, we defined a core outbreak hotspot of Sana’a, Sana’a City, and Al-Hudaydah, which was the expected origin of the national outbreak [42,43,44,45]. These governorates shared nearly identical features as the national outbreak signature in all four waves, including onset and peak timing critical points, peak rate magnitudes, and durations of acceleration and deceleration periods. While key informant interviews and trend analyses elsewhere suggested that Wave 1 peak timing occurred in 5–31 December 2016, we identified earlier peak timing for the core cluster and nationally ranging from 10 October 2016 to 13 November 2016 [26,27,46]. Additionally, various sources including the WHO declared that Wave 1 concluded for all governorates and nationally in 13–27 April 2017 [26,27,46,47]. However, we identified a resolution period for all affected governorates ranging from 2 January 2017 to 5 March 2017.

Our results for Wave 2 outbreak signatures closely aligned with those of Dureab et al. and Camacho et al., whose descriptive trend analyses were based on the Yemeni National Electronic Disease Early Warning System (eDEWS) [26,27]. While Camacho et al. reported a Wave 2 peak at the end of June 2017, Dureab et al. reported a first outbreak peak in 17–23 July 2017 and a second peak of lesser magnitude 3 months later in 18–24 September 2017 [26,27]. We identified two outbreak peaks with similar timing (24–30 July 2017 and 25 September 2017 to 01 October 2017), shared by most governorates. However, we also identified a third peak found in the two neighboring clusters (Amran, Al-Mahwit, Dhamar, Al-Bayda, Raymah, Ibb, and Taizz) surrounding the core outbreak cluster (Sana’a, Sana’a City, and Al-Hudaydah) from 5 March 2018 to 1 April 2018. Similarities in our findings suggested consistency in the reporting of district-level and governorate-level data.

These findings potentially suggested traveling waves from the core outbreak cluster, whose earlier second peak timing (18 September 2017 to 1 October 2017) preceded the later peak timing (1 January 2018 to 20 May 2018) in neighboring governorates. The core cluster reflected a persistent cluster of infections due to consistently high peak rates, early peak timing, and large population density in these governorates. The later peak timing of governorates within close spatial proximity suggested a high degree of percolation to other clusters, though these variations may have been dictated by differences in the spatial distribution of conflict, environmental, and socioeconomic factors within each governorate. Further research is needed to describe patterns of disease spread and differences in drivers of infection in each governorate.

Our findings also suggest a possible second traveling wave from the first neighboring cluster (Amran, Al-Mahwit, Dhamar, and Al-Bayda) to the northern, southern, and eastern remote clusters. The four governorates in this neighboring cluster bordered governorates in each of the three remote clusters. As described earlier, this neighboring cluster is characterized by extremely high peak rate magnitudes (216.3–397.7 cph) for the first peak timing (24 July 2017 to 27 August 2017) in Wave 2 with equally high paces of increase to the peak (10.8–20.9 cph/week). In contrast, most remote clusters had a single peak whose timing was slightly later than the neighboring cluster (24 July 2017 to 17 September 2017) and whose magnitude was lower (22.14–164.88 cph) in most governorates. This suggested that larger outbreaks in this neighboring cluster could have resulted in percolation of the outbreak to the remote clusters, which experienced a short, intense Wave 2 outbreak before quickly returning to pre-outbreak rate values (<4 cph) from 18 December 2017 to 1 April 2018.

While WHO epidemiological surveillance bulletins suggested that Wave 2 persisted from April 2017 through January 2020, we identified two national outbreak waves with onsets in May 2018 and January 2019 [47]. These waves suggest that the epidemic remains ongoing with numerous onsets of outbreak events rather than one prolonged outbreak period. For Waves 1 and 2, the acceleration period from the onset to the peak was 1.5–2.2-fold shorter than the deceleration period from the peak timing to the onset of the next wave. Furthermore, for Wave 3, the acceleration period from the onset to peak was >1.5-fold longer than the deceleration period from the peak of Wave 3 to the onset of Wave 4. As with Waves 1 and 2, we found great heterogeneity in the onset, peak timing, and duration of outbreaks across governorates for Wave 3 and Wave 4. Differences in patterns of outbreak signatures across waves may suggest that drivers of disease transmission varied over time and location. These differences must be inspected individually for each wave and may have required different types of humanitarian relief according to the conflict-, environmental-, and socioeconomic-related drivers of infection.

Discrepancies in outbreak signatures can significantly influence reported associations between cholera rates and human-made or environmental drivers of transmission. For example, Wave 3 began nationally from 28 May 2018 to 3 June 2018, though onset timing ranged from 2 April 2018 to 8 July 2018. The Wave 3 onset timing for southern cluster governorates closely aligned with the arrival of the Mekunu cyclone, which made landfall in Oman and Yemen on 25 May 2018 and resulted in severe flooding [48,49,50]. In contrast, the Battle of Al-Hudaydah from 13 to 22 June 2018 left governorates in the core, neighboring, and remote northern clusters with severe health infrastructure destruction [51]. These natural disaster and conflict events, whose repercussions persisted through December 2019, may explain the shorter acceleration periods and more frequent outbreak peaks in Waves 3 and 4. Future research using our defined outbreak signatures is needed to examine how environmental and human-made risk factors varied both across and within outbreak clusters on a wave-by-wave basis.

Our study could shed light on previous research examining associations between cholera incidence and environmental risk factors for Wave 2 of the Yemeni cholera outbreak. Camacho et al. used district-level surveillance records and high-resolution rainfall data to show that within 10 days of a 10 mm rainfall event the likelihood of cholera infection increased by 21% compared to a week of no rainfall [26]. However, results assumed that each district followed identical outbreak onset and peak timing according to national-level, WHO-defined outbreak wave critical points [26]. Other studies aimed at modeling cholera outbreak signatures similarly did so with pre-defined national-level characteristics, preventing exploration of regional hotspots [27,46,52]. Our results provided more refined estimates of critical points and helped account for the variability of outbreak signatures or differences in outbreak duration across governorates, which faced differing intensities of wartime conflicts, food insecurity, and malnutrition. If more spatially granular time series data were reported at the district level, we could further refine how governorates were assigned to outbreak clusters.

Our study had several limitations. Firstly, the reliability of case data may be impacted by the scattered dispersal of public technical reports and their inconsistent, infrequent, and sometimes non-overlapping time periods. Furthermore, data collection and reporting were expected to be limited by health care services/capacity, utilization of health services, and reporting ability in combination with the ruralness and governing stability of each governorate. The WHO provided no metadata explaining limitations in data quality and coverage after data were aggregated from district-level eDEWS reports to governorate-level WHO bulletins. Dureab et al., who used eDEWS data reports, found only ~1 week lag between the time of infection and reporting of time series records suggesting consistent reporting despite ongoing conflict events [27]. However, key informant interviews in Yemen noted significant destruction of emergency health facilities in Sana’a City and low capacity for routine and reliable laboratory culture testing in Sana’a and Al-Hudaydah [3,5,46]. This suggests possible under-reporting in these governorates, especially during the second wave of the epidemic between March 2017 and April 2018.

We compiled data presented in various formats to the best of our knowledge and abilities and are unaware of alternative methods that could have better validated the compiled data, given the inconsistent reporting and availability of case reports. Similarly, data were only available at the governorate level, which prevented more refined cluster assignments that would account for district-level differences in topography, climate, health infrastructure, etc. While district-level data would improve the granularity of classified outbreak clusters, our use of governorate-level data reflected both the usability of publicly reported data and the reality of the data landscape in Yemen. That said, we found similar critical point timing and magnitudes in our national outbreak signature to the study performed by Dureab et al. (which used eDEWS data), which suggested that our estimation techniques retained the features of reported data [27].

Secondly, population estimates used to calculate weekly rates only accounted for possible changes in population growth rate and conflict fatalities. This growth rate was retrieved from the 2004 Yemeni Census as no additional population estimates or growth rates were provided in the 2014 Census [35,53]. ACLED fatality data used to adjust for violent and non-violent political events were retrieved from a variety of media sources and may not be reliable [54,55,56,57]. Additionally, we were unable to adjust for possible population migration from Yemen or internal displacement within Yemen, as these data were not reported consistently at the monthly or weekly levels [36]. While a substantial proportion (~15%) of Yemen’s population has been internally displaced throughout the Yemeni Civil War, neither ACLED nor any other dataset we explored provided data on migration or internal displacement with sufficient temporal resolution, spatial granularity, or consistent reporting to use in adjusting population estimates. Furthermore, data lacked sufficient spatial resolution to consider examining rates by population density, as most governorates included both urban cities and rural mountainous areas. Because we used a prorated weekly population estimate to account for possible population growth over time in the absence of more refined population data, we likely underestimated the rate of infection.

Finally, data from this study were reported through 29 December 2019. We were unable to extend our time series through 2020 or 2021 due to a lack of epidemiological bulletins for this year [29,47]. Understandably, resources could have been diverted to mitigate and manage the ongoing SARS-CoV-2 pandemic. While weekly updates were provided on the current cholera crisis in Yemen, these reports provided insufficient temporal and spatial granularity for recreating weekly time series of cases. However, various reports suggested, as found in our study, that the cholera epidemic within Yemen remains ongoing with rates like those reported in 2019 [58,59,60,61,62]. Thus, coordinated efforts for mitigating and preventing the spread of cholera are still needed [62,63,64,65,66].

While we detected potential traveling waves for governate-specific outbreaks, there could be many reasons for observed spatiotemporal patterns. The differences in outbreak signatures could be governed by variations in environmental factors, such as rainfall, flooding, and water contamination, conflicts and social unrest, depleted health infrastructure, food insecurity, and many other factors that deserve further investigations. Any such investigation should start with better understanding the spatiotemporal behaviors of infection across clusters and synchronicity between exposure events and outbreaks, that could be explored with dynamic mapping. Dynamic or animated maps are an emerging data visualization technique that compresses large amounts of spatially and temporally aligned records into visuals. We have demonstrated the utility of this visualization technique to identify persistent and percolating clusters of *Salmonella* outbreaks and traveling waves of influenza in the United States [18,67]. Our curated dataset provides the foundation for aligning and integrating other spatiotemporal data on conflict-, environmental-, and socioeconomic-related factors to conduct these future analyses. This expanded dataset can then be used for evaluating the drivers and consequences of disease outbreaks during the Yemeni Civil War.

To increase the usability and transparency of valuable epidemiological records collected by many national and international organizations, public access and dissemination of information are critical. Publicly available data and information could assist in preventing or mitigating humanitarian crises such as this one in Yemen. We encourage the holders of epidemiological records to share data and metadata with the broader scientific community at the highest temporal and spatial granularity possible to ease the extraction, understanding, and utilization of publicly disseminated data. We also encourage future applications of the novel methods for early outbreak detection, regional hotspot identification, and resource distribution coordination in accordance with the global mandate of the WHO’s GTFCC [8,9,10,11]. Our findings, in conjunction with future research, could support the GTFCC roadmap and provide solid methodology to define outbreak signatures, classify regional hotspots, evaluate risk factors contributing to outbreak transmission, and establish an early outbreak warning system [10,11].

## 5. Conclusions

Our study provided a data-driven approach to describe and compare outbreak signatures of governorate-level cholera outbreaks in Yemen. These techniques utilized time series surveillance data and were not unique to the temporal resolution, spatial granularity, or infectious disease examined. Though our analysis evaluated data retrospectively, the proposed methodology can be adapted and applied for real-time forecasting to determine outbreak signatures of ongoing epidemics. Furthermore, our approach demonstrated how data published publicly by humanitarian organizations can be retrieved, harmonized, and utilized for modeling outbreak signatures. The detected heterogeneity in outbreak signatures associated with spatial proximity suggested possible traveling waves of infection outwards from the core cluster. Further research is needed to investigate if these differences are due to impacts of conflict, environmental, social, and economic factors across governorates. Similarities in outbreak signatures also suggested potentially shared environmental and human-made drivers of infection. By identifying and grouping governorates with similar critical points, periods, and rate magnitudes, we provided the context for identifying and comparing human-made and environmental drivers of disease transmission. These methods can help to parse out associations between infectious disease outbreaks and wartime conflict, rainfall, food insecurity, and consumer price risk factors to assist in coordinating humanitarian relief efforts.

## Figures and Tables

**Figure 1 ijerph-19-00378-f001:**
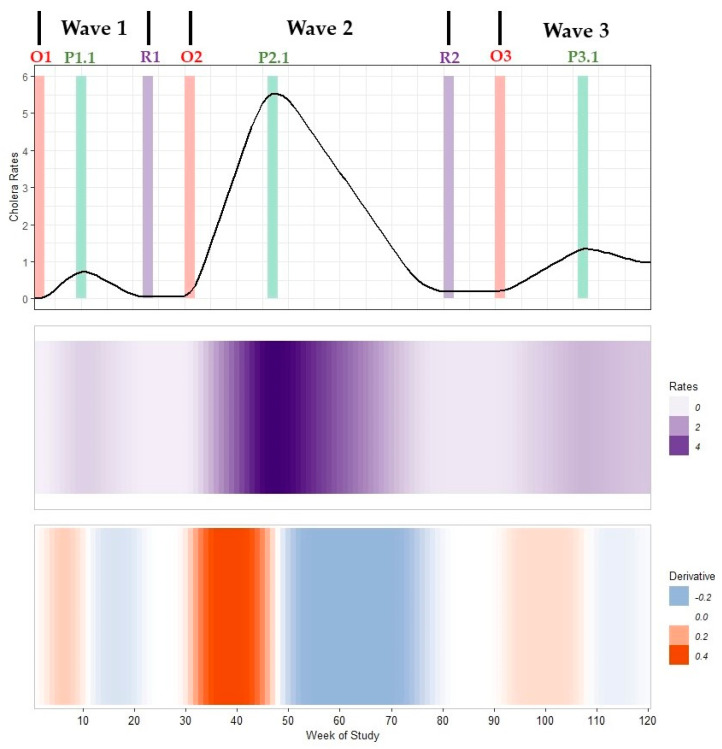
A multi-panel plot of an example cholera outbreak signature with three waves. Top panel: a time series of smoothed average rates of cholera infection (an epidemic curve). Middle panel: a heatmap that compactly reports weekly rates (outbreak magnitude) where light purple represents low values and dark purple represents high values. Bottom panel: a heatmap of the change in rates (trivial derivative) where acceleration periods are shown by the orange gradient, deceleration periods are shown by the blue gradient, and steady-state periods or critical change points near peaks are shown in grey. All panels share the common horizontal axis of time (study week). The timing of outbreak onset (*O_n_*), peak (*P_n.i_*), and resolution (*R_n_*) are marked with red, green and purple bars, respectively, for the *n*th wave and the *i*th peak within that wave. If no resolution period occurs, the outbreak remains ongoing until reaching resolution or the next wave’s onset (shown in Wave 3). Data used to develop this visualization are reported in Appendix A.

**Figure 2 ijerph-19-00378-f002:**
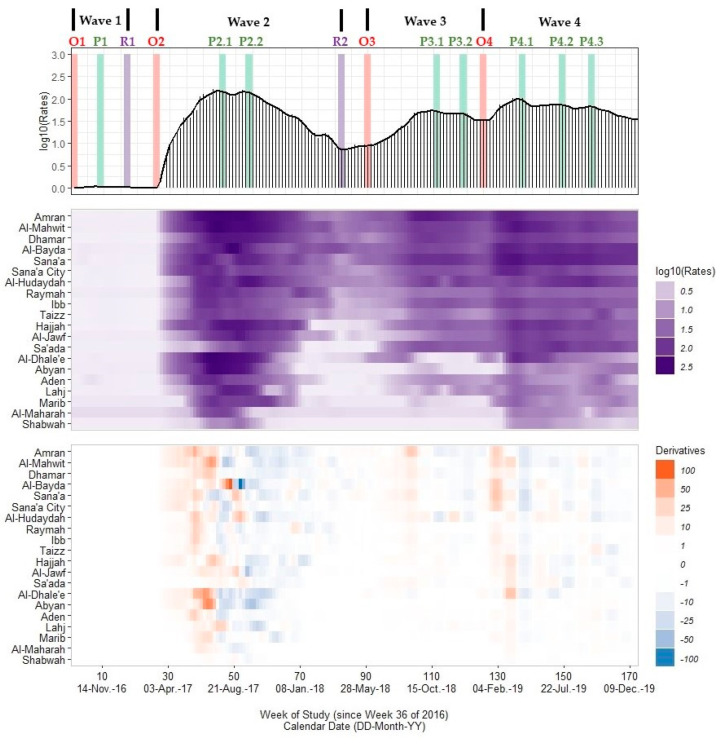
A multi-panel plot of the national Yemeni cholera outbreak signature (top panel), a heatmap of rate values across 20 governorates (middle panel; excludes the Hadramaut and Socotra island governorate), and a heatmap of differences in rates or trivial derivative values (bottom panel). Cholera rates (cases per 100,000 persons, or cph) are presented as a spike plot using a log10-transformed scale for 172 study weeks (all missing data imputed with linear approximations). A smoothed time series line plot is superimposed on this spike plot, which was calculated using Kolmogorov–Zurbenko adaptive filters from Week 36 (4–11 September) of 2016 through Week 52 (23–29 December) of 2019. Smoothed rates (laboratory-confirmed cholera cases per 100,000 persons) were estimated using the best performing smoother. Vertical bars represent the onset timing (red) for each wave (O1, O2, O3, O4), the peak timing (green) including one peak for Wave 1 (P1), two peaks for Wave 2 (P2.1, P2.2) and Wave 3 (P3.1, P3.2), and three peaks for Wave 4 (P4.1, P4.2, P4.3), and the resolution timing (purple) for Waves 1 and 2 (R1, R2). The heatmap of rates uses a purple gradient scale (lighter hues ~0.0 cph and darker hues ~400 cph) while the heatmap of derivative values uses a diverging scale ranging from −100 (dark blue) to +100 (dark orange) with values near zero in grey. All plots share a common horizontal axis of time reported in weeks since Week 36 of 2016 (defined as study Week 0). Governorates are grouped by assigned outbreak clusters: the core outbreak (Sana’a, Sana’a City, and Al-Hudaydah), immediate neighboring I (Amran, Al-Mahwit, Dhamar, and Al-Bayda), immediate neighboring II (Raymah, Ibb, and Taizz), northern (Hajjah, Al-Jawf, and Sa’ada), southern (Al-Dhale’e, Abyan, Aden, and Lahj), and eastern (Marib, Al-Maharah, and Shabwah) clusters. Data used to develop this visualization are reported in Table 1 and Table 2 and Appendix A.

**Figure 3 ijerph-19-00378-f003:**
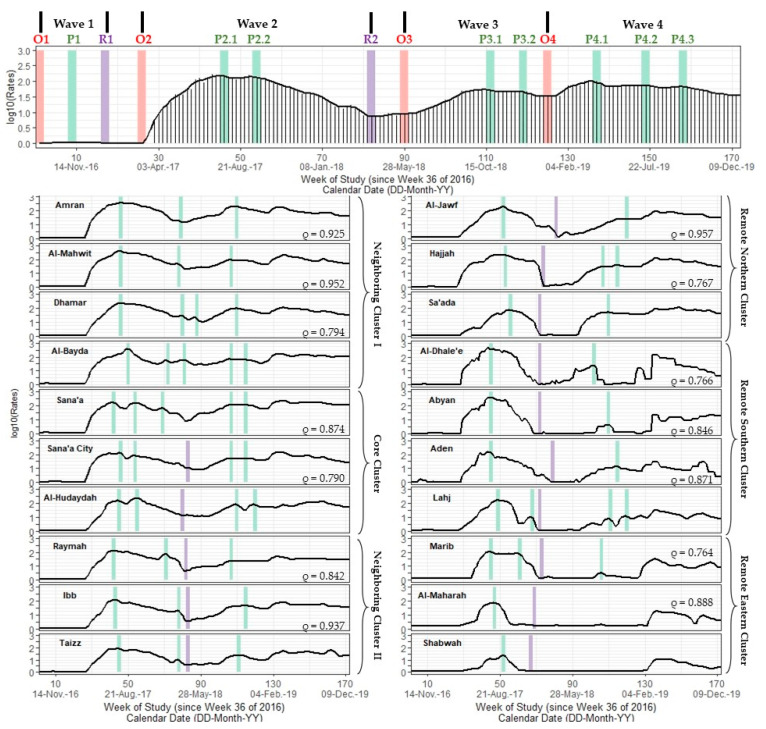
A multi-panel plot of the national Yemeni cholera outbreak signature and signatures for 20 Yemeni governorates (excludes the Hadramaut and Socotra island governorate). Top panel: the national signature includes a spike plot of log10-transformed cholera rates (in cases per 100,000 persons) and a time series line plot of smoothed rates calculated using Kolmogorov–Zurbenko adaptive filters from Week 36 (4–11 September) of 2016 through Week 52 (23–29 December) of 2019. Smoothed rates (laboratory-confirmed cholera cases per 100,000 persons) were estimated using the best performing smoother. Vertical red, green, and purple lines depict the timing of onset, peak, and resolution critical point (s), respectively, for each of the four outbreak waves. Bottom panels: governorate-level signatures are shown in two columns of stacked multi-panel time series plots that share a common vertical axis between columns (log10-transformed rates (cph)) and a common horizontal axis within columns (weeks since Week 36 of 2016, defined as study Week 0). Vertical green and purple bars depict the timing of the Wave 2 peak, Wave 2 resolution, and Wave 3 peak, which were important signature features for defining outbreak clusters. Spearman cross correlation estimates (ρ) are at lag 0 between adjacently reported governorates within the same cluster based on 172 weeks (all correlation coefficients are significant at α < 0.001). Data used to develop this visualization are reported in Table 1, Appendix A.

**Figure 4 ijerph-19-00378-f004:**
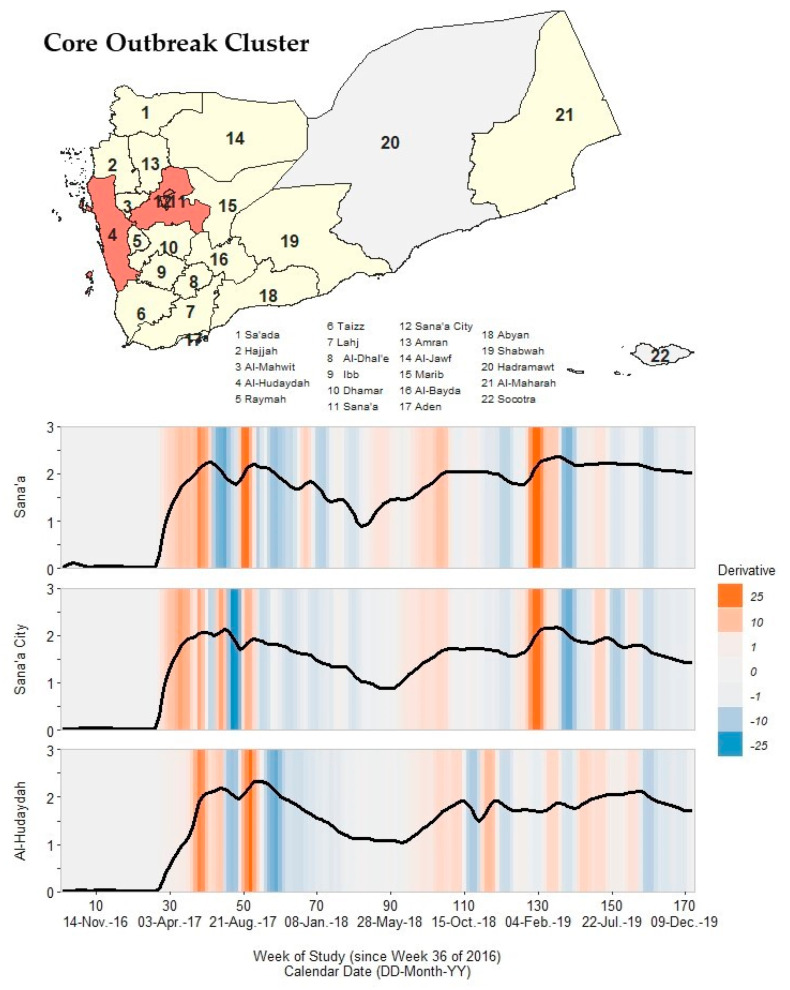
A map of Yemen with a shaded outbreak cluster including Sana’a, Sana’a City, and Al-Hudaydah, and a stacked time series plot and heatmap for each governorate’s outbreak signature. Top panel: the governorate-level map indicates governorates within the cluster in orange, remaining governorates within our study in yellow, and excluded governorates in grey (the Hadramaut and Socotra island governorate). Bottom panel: stacked time series plots for each governorate within the cluster. All plots show log10-transformed smoothed rates produced using Kolmogorov–Zurbenko adaptive filters from Week 36 (4–11 September) of 2016 through Week 52 (23–29 December) of 2019. Smoothed rates (laboratory-confirmed cholera cases per 100,000 persons) were estimated using the best performing smoother. Backgrounds for each plot are heatmaps of trivial derivative values, ranging from −25 (dark blue) to +25 (dark orange) with values near zero in grey. Time series plots have a common horizontal axis of time reported in weeks since Week 36 of 2016 (defined as study Week 0). Data used to develop this visualization are reported in Table 1 and Appendix A.

**Figure 5 ijerph-19-00378-f005:**
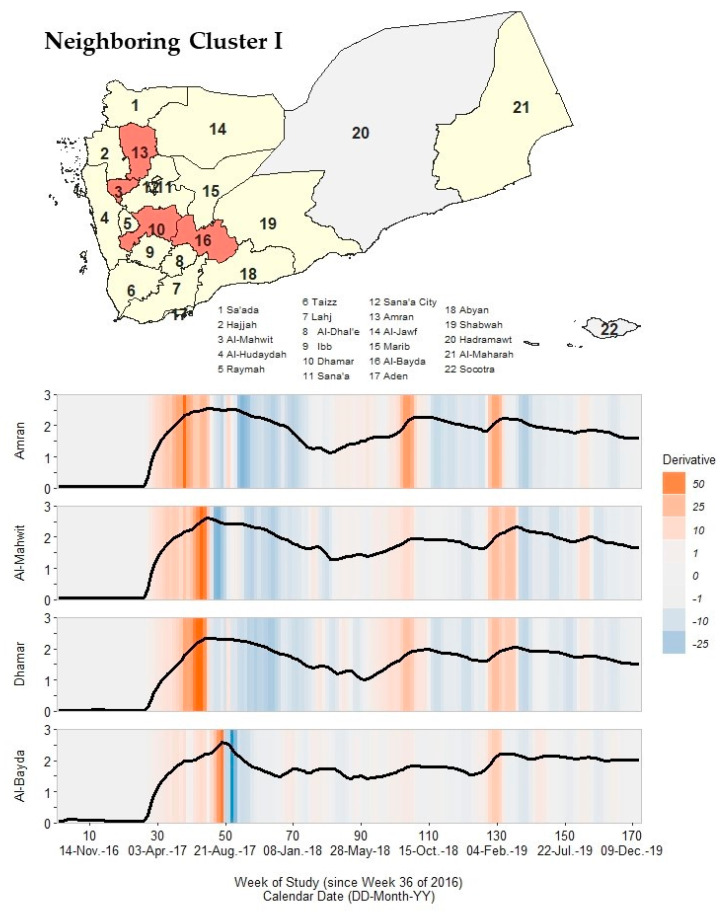
A map of Yemen with a shaded outbreak cluster including Amran, Al-Mahwit, Dhamar, and Al-Bayda, and a stacked time series plot and heatmap for each governorate’s outbreak signature. Top panel: the governorate-level map indicates governorates within the cluster in orange, remaining governorates within our study in yellow, and excluded governorates in grey (the Hadramaut and Socotra island governorate). Bottom panel: stacked time series plots for each governorate within the cluster. All plots show log10-transformed smoothed rates produced using Kolmogorov–Zurbenko adaptive filters from Week 36 (4–11 September) of 2016 through Week 52 (23–29 December) of 2019. Smoothed rates (laboratory-confirmed cholera cases per 100,000 persons) were estimated using the best performing smoother. Backgrounds for each plot are heatmaps of trivial derivative values, ranging from −25 (dark blue) to +50 (dark orange) with values near zero in grey. Time series plots have a common horizontal axis of time reported in weeks since Week 36 of 2016 (defined as study Week 0). All rates and trivial derivates were calculated for the 172 weeks of time series. Data used to develop this visualization are reported in Table 1 and Appendix A.

**Figure 6 ijerph-19-00378-f006:**
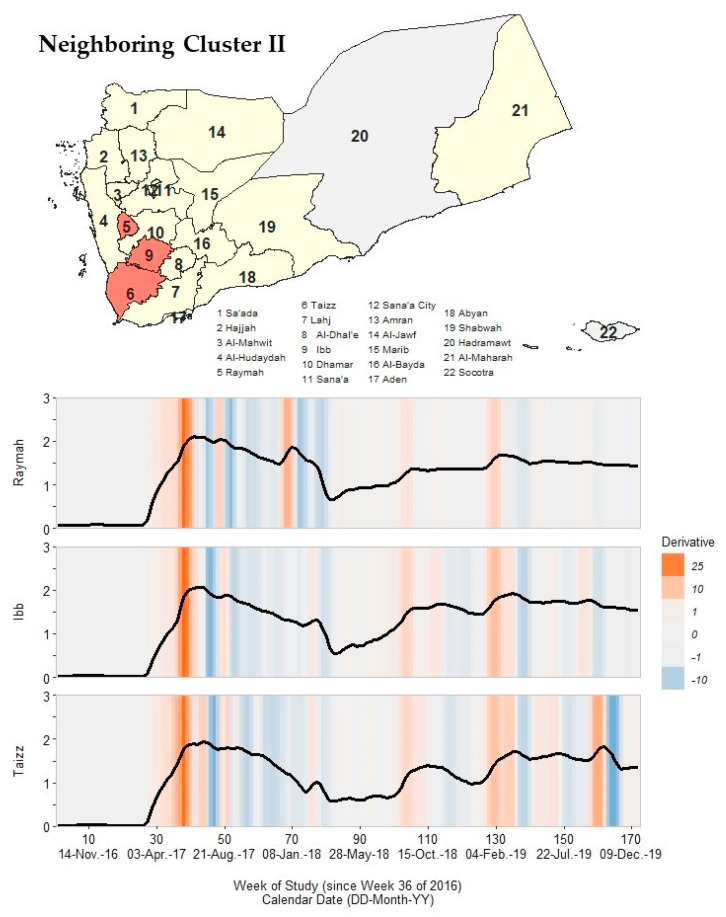
A map of Yemen with a shaded outbreak cluster including Raymah, Ibb, and Taizz, and a stacked time series plot and heatmap for each governorate’s outbreak signature. Top panel: the governorate-level map indicates governorates within the cluster in orange, remaining governorates within our study in yellow, and excluded governorates in grey (the Hadramaut and Socotra island governorate). Bottom panel: stacked time series plots for each governorate within the cluster. All plots show log10-transformed smoothed rates produced using Kolmogorov–Zurbenko adaptive filters from Week 36 (4–11 September) of 2016 through Week 52 (23–29 December) of 2019. Smoothed rates (laboratory-confirmed cholera cases per 100,000 persons) were estimated using the best performing smoother. Backgrounds for each plot are heatmaps of trivial derivative values, ranging from −10 (dark blue) to +25 (dark orange) with values near zero in grey. Time series plots have a common horizontal axis of time reported in weeks since Week 36 of 2016 (defined as study Week 0). All rates and trivial derivates were calculated for the 172 weeks of time series. Data used to develop this visualization are reported in Table 1 and Appendix A.

**Figure 7 ijerph-19-00378-f007:**
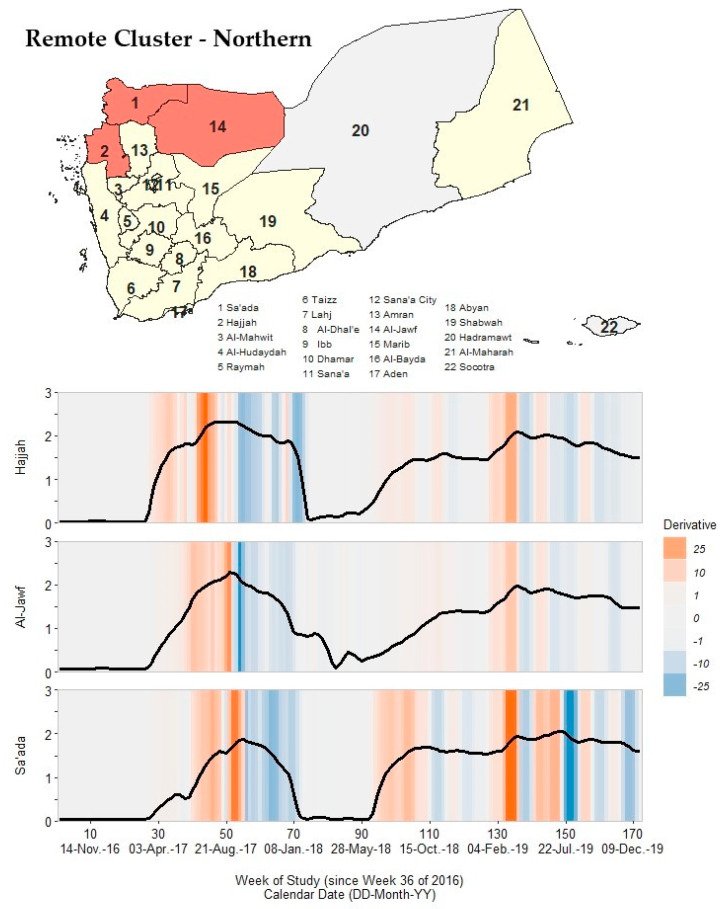
A map of Yemen with a shaded outbreak cluster including Hajjah, Al-Jawf, and Sa’ada, and a stacked time series plot and heatmap for each governorate’s outbreak signature. Top panel: the governorate-level map indicates governorates within the cluster in orange, remaining governorates within our study in yellow, and excluded governorates in grey (the Hadramaut and Socotra island governorate). Bottom panel: stacked time series plots for each governorate within the cluster. All plots show log10-transformed smoothed rates produced using Kolmogorov–Zurbenko adaptive filters from Week 36 (4–11 September) of 2016 through Week 52 (23–29 December) of 2019. Smoothed rates (laboratory-confirmed cholera cases per 100,000 persons) were estimated using the best performing smoother. Backgrounds for each plot are heatmaps of trivial derivative values, ranging from −25 (dark blue) to +25 (dark orange) with values near zero in grey. Time series plots have a common horizontal axis of time reported in weeks since Week 36 of 2016 (defined as study Week 0). All rates and trivial derivates were calculated for the 172 weeks of time series. Data used to develop this visualization are reported in Table 1 and Appendix A.

**Figure 8 ijerph-19-00378-f008:**
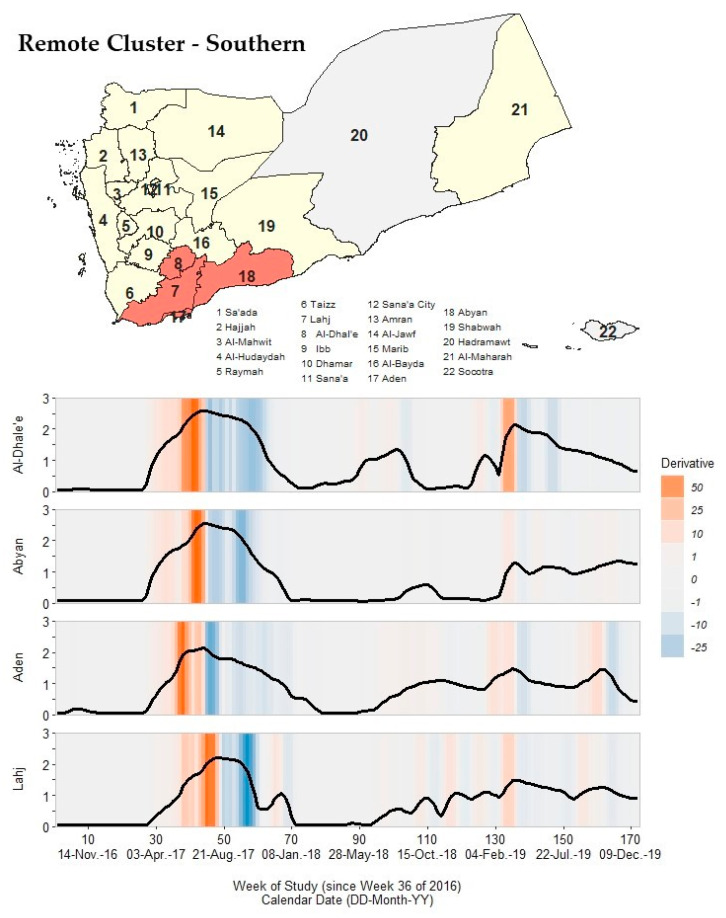
A map of Yemen with a shaded outbreak cluster including Al-Dhale’e, Abyan, Aden, and Lahj, and a stacked time series plot and heatmap for each governorate’s outbreak signature. Top panel: the governorate-level map indicates governorates within the cluster in orange, remaining governorates within our study in yellow, and excluded governorates in grey (the Hadramaut and Socotra island governorate). Bottom panel: stacked time series plots for each governorate within the cluster. All plots show log10-transformed smoothed rates produced using Kolmogorov–Zurbenko adaptive filters from Week 36 (4–11 September) of 2016 through Week 52 (23–29 December) of 2019. Smoothed rates (laboratory-confirmed cholera cases per 100,000 persons) were estimated using the best performing smoother. Backgrounds for each plot are heatmaps of trivial derivative values, ranging from −25 (dark blue) to +50 (dark orange) with values near zero in grey. Time series plots have a common horizontal axis of time reported in weeks since Week 36 of 2016 (defined as study Week 0). All rates and trivial derivates were calculated for the 172 weeks of time series. Data used to develop this visualization are reported in Table 1 and Appendix A.

**Figure 9 ijerph-19-00378-f009:**
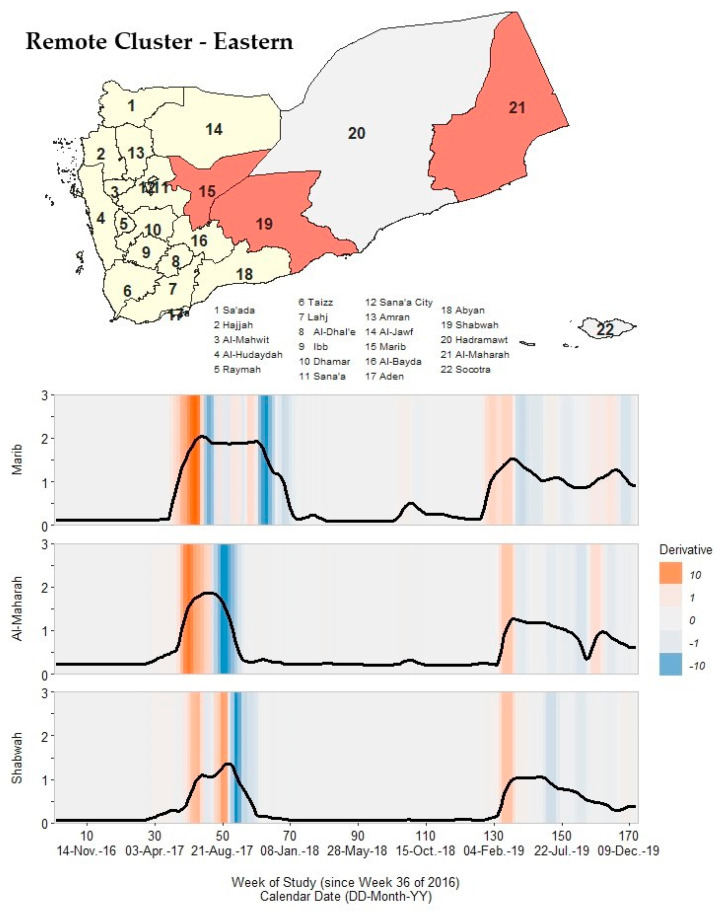
A map of Yemen with a shaded outbreak cluster including Marib, Al-Maharah, and Shabwah, and a stacked time series plot and heatmap for each governorate’s outbreak signature. Top panel: the governorate-level map indicates governorates within the cluster in orange, remaining governorates within our study in yellow, and excluded governorates in grey (the Hadramaut and Socotra island governorate). Bottom panel: stacked time series plots for each governorate within the cluster. All plots show log10-transformed smoothed rates produced using Kolmogorov–Zurbenko adaptive filters from Week 36 (4–11 September) of 2016 through Week 52 (23–29 December) of 2019. Smoothed rates (laboratory-confirmed cholera cases per 100,000 persons) were estimated using the best performing smoother. Backgrounds for each plot are heatmaps of trivial derivative values, ranging from −10 (dark blue) to +10 (dark orange) with values near zero in grey. Time series plots have a common horizontal axis of time reported in weeks since Week 36 of 2016 (defined as study Week 0). All rates and trivial derivates were calculated for the 172 weeks of time series. Data used to develop this visualization are reported in Table 1 and Appendix A.

**Table 1 ijerph-19-00378-t001:** Timing and magnitude of smoothed rate values at the onset, peak, and resolution critical points for four outbreak waves in Yemen from Week 36 (4–11 September) of 2016 through Week 52 (23–29 December) of 2019.

Location	Metric	O1	P1	R1	O2.1	P2.1	O2.2	P2.2	O2.3	P2.3	R2	O3.1	P3.1	O3.2	P3.2	R3	O4.1	P4.1	O4.2	P4.2	O4.3	P4.3
Yemen	Timing	1	9	17	26	46	50	54			82	90	111	116	119		125	137	142	149	154	158
Yemen	Rate	0.01	0.06	0.03	0.01	151.33	122.17	141.13			6.34	7.89	53.12	44.42	45.59		31.84	99.65	66.79	73.41	59.91	66.08
Sana’a	Timing	1	5	17	26	42	49	54	65	69		83	107	111	115		127	136	142	150		
Sana’a	Rate	0.07	0.29	0.10	0.08	172.81	58.29	154.88	47.06	66.78		6.71	111.35	108.16	112.59		58.12	228.81	150.93	170.86		
Sana’a City	Timing	3	9	17	26	46	50	54			83	92	107	111	115		124	136	145	150	155	158
Sana’a City	Rate	0.03	0.14	0.06	0.04	131.36	49.80	82.07			9.23	6.82	50.96	50.24	51.07		34.71	146.37	67.45	86.48	54.42	61.33
Al-Hudaydah	Timing	3	7	17	26	45	50	55			80	94	110	115	120		132	137	141	158		
Al-Hudaydah	Rate	0.03	0.10	0.06	0.04	157.26	91.28	215.79			12.29	9.87	81.55	29.26	81.77		47.25	74.25	56.16	131.58		
Amran	Timing	5	13	17	26	46	77	79				82	110				128	133	154	159		
Amran	Rate	0.09	0.16	0.12	0.09	347.50	17.30	17.97				12.67	188.36				65.22	166.90	58.39	70.71		
Al-Mahwit	Timing				27	46	75	78				83	107				127	137	154	159		
Al-Mahwit	Rate				0.14	397.67	42.16	49.68				18.50	90.84				41.36	208.86	70.65	99.38		
Dhamar	Timing	8	13	17	26	46	77	80	84	88		92	110				127	137	155	158		
Dhamar	Rate	0.05	0.12	0.08	0.05	216.32	22.95	27.49	14.35	19.02		9.44	93.51				38.23	112.44	53.09	58.28		
Al-Bayda	Timing	2	6	17	26	50	67	72	75	81		93	107	111	115		124	133	141	146	154	158
Al-Bayda	Rate	0.13	0.28	0.16	0.14	391.53	29.84	52.82	39.38	54.40		25.95	68.28	60.67	63.07		33.13	166.26	108.80	135.80	112.76	125.31
Raymah	Timing	8	13	17	26	42	67	71			82	98	107				124	133	141	146	154	158
Raymah	Rate	0.17	0.24	0.19	0.17	125.58	28.60	70.79			3.56	8.59	22.26				21.98	48.00	29.76	33.78	30.44	32.10
Ibb	Timing	3	9	17	26	43	74	78			83	90	115				125	136	145	150	154	158
Ibb	Rate	0.04	0.12	0.06	0.04	113.44	14.45	19.22			2.45	4.12	47.16				27.81	81.19	50.29	54.41	48.52	57.77
Taizz	Timing	3	10	17	26	45	75	78			83	99	111				124	137	141	150	158	163
Taizz	Rate	0.03	0.12	0.06	0.04	85.69	5.04	9.25			2.92	3.54	23.40				8.40	51.69	33.98	45.95	30.86	65.77
Hajjah	Timing	3	13	17	26	53					74	90	107	111	115		127	137	141	145	154	158
Hajjah	Rate	0.05	0.12	0.07	0.05	208.08					0.24	0.58	38.23	0.17	0.41		27.33	121.84	89.32	103.53	59.54	69.61
Al-Jawf	Timing	10	15	17	26	52					81	91	120				127	137	141	145	155	159
Al-Jawf	Rate	0.17	0.22	0.20	0.18	192.51					0.77	0.78	23.95				22.20	91.12	65.35	80.27	49.93	53.51
Sa’ada	Timing				27	56					72	92	110				128	137	141	149	155	158
Sa’ada	Rate				0.10	72.96					0.19	0.10	48.94				33.86	87.60	72.49	111.12	64.63	72.69
Al-Dhale’e	Timing	5	9	13	27	45					72	95	102			111	123	128	132	137		
Al-Dhale’e	Rate	0.14	0.25	0.14	0.14	394.66					0.32	9.53	20.62			0.17	0.28	13.08	2.31	133.76		
Abyan	Timing	8	13	17	26	45					72	93	110			115	128	137	141	146	154	167
Abyan	Rate	0.18	0.25	0.20	0.18	358.26					0.17	0.17	2.96			0.40	0.20	18.60	7.71	13.60	7.67	21.13
Aden	Timing	3	8	17	26	45					79	94	115				127	136	154	163		
Aden	Rate	0.11	0.56	0.14	0.11	134.95					0.15	0.39	11.69				5.32	28.10	6.40	27.46		
Lahj	Timing	3	9	17	26	49	64	68			72	93	111	115	120		124	128	132	137	154	163
Lahj	Rate	0.10	0.17	0.13	0.10	164.88	2.63	10.37			0.13	0.10	7.26	1.13	10.99		6.56	11.85	7.85	30.16	6.83	17.10
Marib	Timing				27	45	58	61			73	101	106				124	137	146	149	155	167
Marib	Rate				0.29	110.22	75.11	83.20			0.41	0.28	2.14				0.40	33.00	9.34	11.47	6.43	17.72
Al-Maharah	Timing				27	47					69						132	137	158	163		
Al-Maharah	Rate				0.66	72.63					0.65						0.63	17.68	1.24	8.48		
Shabwah	Timing				27	52					67						127	145				
Shabwah	Rate				0.16	22.14					0.26						0.16	10.21				

All critical points were estimated using smoothed rates (laboratory-confirmed cholera cases per 100,000 persons) and trivial derivative values (Δ rates per epidemiological week) produced with Kolmogorov–Zurbenko (KZ) adaptive filters. We selected the best performing smoother ($Z_{t,s,3\text{–}5}^{*}$) as the average of a 3 week ($Z_{t+1,s,3}^{*}$) and 5 week ($Z_{t+1,s,5}^{*}$) window size and trivial derivative as the absolute change in weekly smoothed rates calculated as: $Z_{t,s,3\text{–}5}^{\prime }=\left(Z_{t+1,s,3}^{*}+Z_{t+1,s,5}^{*}\right)/2-\left(Z_{t,s,3}^{*}+Z_{t,s,5}^{*}\right)/2$. Onset (*O_n.i_*) critical points marked a rapid rise in $Z_{t,s,3\text{–}5}^{*}$ and near-zero to high positive values of $Z_{t,s,3\text{–}5}^{\prime }$. Peak (*P_n.i_*) critical points marked local maximum of $Z_{t,s,3\text{–}5}^{*}$ and inflection of $Z_{t,s,3\text{–}5}^{\prime }$ at near-zero or negative values. Resolution (*R_n_*) critical points defined times when $Z_{t,s,3\text{–}5}^{*}$ were near local minimum values and $Z_{t,s,3\text{–}5}^{\prime }$ approached near-zero values. All critical points are listed for each of the *n*th outbreak waves and *i*th occurrences per wave. No resolution critical point for Wave 4 was found by the conclusion of the study period. Critical points are reported in weeks since Week 36 of 2016 (defined as study Week 0). Smoothed rates are reported for the week of each critical point and were estimated for each governorate and nationally using 172 weeks of data. All weeks with missing data were imputed using linear approximation. Governorates are grouped by assigned outbreak clusters (separated by double solid lines) and include: the core outbreak (Sana’a, Sana’a City, and Al-Hudaydah), immediate neighboring I (Amran, Al-Mahwit, Dhamar, and Al-Bayda), immediate neighboring II (Raymah, Ibb, and Taizz), northern (Hajjah, Al-Jawf, and Sa’ada), southern (Al-Dhale’e, Abyan, Aden, and Lahj), and eastern (Marib, Al-Maharah, and Shabwah) clusters. Data summarized in this table are graphically presented in Figures 2–9 and Appendix A.

**Table 2 ijerph-19-00378-t002:** Summary of the acceleration, deceleration, and steady-state critical period durations and the pace of increase and decline for four outbreak waves in Yemen from Week 36 (4–11-September) of 2016 through Week 52 (23–29-December) of 2019.

	From	O1	P1	P1	R1	O2.1	O2.1	O2.1	P2.1	P2.2	P2.3	P2.1	P2.2	P2.3	R2	O3.1	O3.1	P3.1	P3.1	P3.2	R3	O4.1	O4.1	O4.1
Location	To	P1	R1	O2.1	O2.1	P2.1	P2.2	P2.3	R2	R2	R2	O3.1	O3.1	O3.1	O3.1	P3.1	P3.2	R3	O4.1	O4.1	O4.1	P4.1	P4.2	P4.3
Yemen	Duration	8	8	17	9	20	28		36	28		44	36		8	21	29		14	6		24	29	33
Yemen	Pace	0.01	0.00	0.00	0.00	7.57	5.04		−4.03	−4.81		−3.26	−2.86		0.19	2.15	1.30		−1.52	−2.29		5.65	1.73	1.04
Sana’a	Duration	4	12	21	9	16	28	43				41	29	14		24	32		20	12		9	23	
Sana’a	Pace	0.07	−0.02	−0.01	0.00	10.80	5.53	1.55				−4.05	−5.11	−4.29		4.36	3.31		−2.66	−4.54		18.97	4.90	
Sana’a City	Duration	6	8	17	9	20	28		37	29		46	38		9	15	23		17	9		26	31	34
Sana’a City	Pace	0.03	−0.01	−0.01	0.00	6.57	2.93		−3.30	−2.51		−2.71	−1.98		−0.27	2.94	1.92		−0.96	−1.82		9.31	1.99	0.78
Al-Hudaydah	Duration	4	10	19	9	19	29		35	25		49	39		14	16	26		22	12		5	26	
Al-Hudaydah	Pace	0.03	0.00	0.00	0.00	8.27	7.44		−4.14	−8.14		−3.01	−5.28		−0.17	4.48	2.77		−1.56	−2.88		5.40	3.24	
Amran	Duration	8	4	13	9	20	53					36	3			28			18			5	31	
Amran	Pace	0.09	−0.01	−0.01	0.00	17.37	0.34					−9.30	−1.77			6.27			−6.84			20.34	0.18	
Al-Mahwit	Duration					19	51					37	5			24			20			10	32	
Al-Mahwit	Pace					20.92	0.97					−10.25	−6.24			3.01			−2.47			16.75	1.81	
Dhamar	Duration	5	4	13	9	20	54	62				46	12	4		18			17			10	31	
Dhamar	Pace	0.01	−0.01	−0.01	0.00	10.81	0.51	0.31				−4.50	−1.50	−2.40		4.67			−3.25			7.42	0.65	
Al-Bayda	Duration	4	11	20	9	24	46	55				43	21	12		14	22		17	9		9	22	34
Al-Bayda	Pace	0.04	−0.01	−0.01	0.00	16.31	1.15	0.99				−8.50	−1.28	−2.37		3.02	1.69		−3.25	0.30		14.79	4.67	2.71
Raymah	Duration	5	4	13	9	16	45		40	11		56	27		16	9			17			9	22	34
Raymah	Pace	0.01	−0.01	−0.01	0.00	7.84	1.57		−3.05	−6.11		−2.09	−2.30		0.31	1.52			−0.02			2.89	0.54	0.30
Ibb	Duration	6	8	17	9	17	52		40	5		47	12		7	25			10			11	25	33
Ibb	Pace	0.01	−0.01	0.00	0.00	6.67	0.37		−2.77	−3.35		−2.33	−1.26		0.24	1.72			−1.94			4.85	1.06	0.91
Taizz	Duration	7	7	16	9	19	52		38	5		54	21		16	12			13			13	26	39
Taizz	Pace	0.01	−0.01	−0.01	0.00	4.51	0.18		−2.18	−1.27		−1.52	−0.27		0.04	1.66			−1.15			3.33	1.44	1.47
Hajjah	Duration	10	4	13	9	27			21			37			16	17	25		20	12		10	18	31
Hajjah	Pace	0.01	−0.01	−0.01	0.00	7.70			−9.90			−5.61			0.02	2.21	−0.01		−0.55	2.24		9.45	4.23	1.36
Al-Jawf	Duration	5	2	11	9	26			29			39			10	29			7			10	18	32
Al-Jawf	Pace	0.01	−0.01	0.00	0.00	7.40			−6.61			−4.92			0.00	0.80			−0.25			6.89	3.23	0.98
Sa’ada	Duration					29			16			36			20	18			18			9	21	30
Sa’ada	Pace					2.51			−4.55			−2.02			0.00	2.71			−0.84			5.97	3.68	1.29
Al-Dhale’e	Duration	4	4	18	14	18			27			50			23	7		9	21		12	5	14	
Al-Dhale’e	Pace	0.03	−0.03	−0.01	0.00	21.92			−14.61			−7.70			0.40	1.58		−2.27	−0.97		0.01	2.56	9.53	
Abyan	Duration	5	4	13	9	19			27			48			21	17		5	18		13	9	18	39
Abyan	Pace	0.01	−0.01	−0.01	0.00	18.85			−13.26			−7.46			0.00	0.16		−0.51	−0.15		−0.02	2.04	0.74	0.54
Aden	Duration	5	9	18	9	19			34			49			15	21			12			9	36	
Aden	Pace	0.09	−0.05	−0.03	0.00	7.10			−3.96			−2.75			0.02	0.54			−0.53			2.53	0.62	
Lahj	Duration	6	8	17	9	23	42		23	4		44	25		21	18	27		13	4		4	13	39
Lahj	Pace	0.01	−0.01	0.00	0.00	7.16	0.24		−7.16	−2.56		−3.75	−0.41		0.00	0.40	0.40		−0.05	−1.11		1.32	1.82	0.27
Marib	Duration					18	34		28	12		56	40		28	5			18			13	25	43
Marib	Pace					6.11	2.44		−3.92	−6.90		−1.96	−2.07		0.00	0.37			−0.10			2.51	0.44	0.40
Al-Maharah	Duration					20			22													5	31	
Al-Maharah	Pace					3.60			−3.27													3.41	0.25	
Shabwah	Duration					25			15													18		
Shabwah	Pace					0.88			−1.46													0.56		

Deceleration periods are reported for both the duration from a wave’s peak to resolution and from a wave’s peak to the following wave’s onset. Acceleration periods indicated times when weekly rates were steadily increasing (onset to peak), deceleration periods indicated times when weekly rates were steadily decreasing (peak to resolution or peak to onset), and steady-state periods indicated times when the weekly rates plateaued (resolution to onset). The pace of increase or decline is measured as a linear slope (rates per week) and estimated simply as the smoothed rate at one critical point minus the smoothed rate at the following critical point and divided by the duration between critical points. Duration between critical points ranged 2–56 weeks. Critical periods are denoted from their starting and ending onset (*O_n.i_*), peak (*P_n.i_*), and resolution (*R_n_*) critical points for the *n*th outbreak wave and *i*th occurrence per wave. No resolution critical point for Wave 4 was found by the conclusion of the study period. As a result, we found no deceleration or steady-state periods for Wave 4. Critical period duration is reported in weeks since Week 36 of 2016 (defined as study Week 0). The pace of increase and decline are reported in laboratory-confirmed cholera cases per 100,000 persons per week. Critical points used to define critical periods were estimated using smoothed rates (laboratory-confirmed cholera cases/100,000 persons) and trivial derivative values (Δ rates/week) produced with Kolmogorov–Zurbenko (KZ) adaptive filters (see Table 1). Governorates are grouped by assigned outbreak clusters indicated with double solid lines and include: the core outbreak (Sana’a, Sana’a City, and Al-Hudaydah), immediate neighboring I (Amran, Al-Mahwit, Dhamar, and Al-Bayda), immediate neighboring II (Raymah, Ibb, and Taizz), northern (Hajjah, Al-Jawf, and Sa’ada), southern (Al-Dhale’e, Abyan, Aden, and Lahj), and eastern (Marib, Al-Maharah, and Shabwah) clusters. Data summarized in this table are, in part, graphically presented in Figure 2 and Appendix A.

## Data Availability

All data and visualization coding resources are reported in Appendix A.

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
