# Peer review of "Signatures of Cholera Outbreak during the Yemeni Civil War, 2016–2019"

_ijerph, 2021, doi:10.3390/ijerph19010378_

Round 1

Reviewer 1 Report

Simpson et al manuscript Signatures of cholera outbreak in Yemeni Civil War, 2016-2019”retrospectively analyse data of cholera positive patients in Yemen collected in 172-week period (2016-2019), during the Yemen civil war. The newest data, considering 2020-2021. were not analysed, but authors have explained the lack of epidemiological bulletins for this year, that is acceptable explanation.

Authors collected raw data of cholera cases that were detected in that period, and they were characterized, classified and compared considering different parameters. For their analysis, authors have applied non-parametric Kolmogorov-Zurbenko adaptive filters and even subtle differences between specific regions in Yemen are well discerned. The results are very nicely and logically presented, both the figures and tables, and it is easy to follow them. It is very interesting to follow the kinetics of the outbreak. The limitations and problems, especially considering data collection are explained and authors give effort to maximally compensate it. The discussion is also well written and analyse achieved results with the results from the literature.

I find that this study contribute to the quality of outbreak analysis, and I suggest the acceptance of the manuscript.

Author Response

Simpson et al manuscript Signatures of cholera outbreak in Yemeni Civil War, 2016-2019” retrospectively analyse data of cholera positive patients in Yemen collected in 172-week period (2016-2019), during the Yemen civil war. The newest data, considering 2020-2021 were not analysed, but authors have explained the lack of epidemiological bulletins for this year, that is acceptable explanation.

We thank the Reviewer.

Authors collected raw data of cholera cases that were detected in that period, and they were characterized, classified and compared considering different parameters. For their analysis, authors have applied non-parametric Kolmogorov-Zurbenko adaptive filters and even subtle differences between specific regions in Yemen are well discerned. The results are very nicely and logically presented, both the figures and tables, and it is easy to follow them. It is very interesting to follow the kinetics of the outbreak. The limitations and problems, especially considering data collection are explained and authors give effort to maximally compensate it. The discussion is also well written and analyse achieved results with the results from the literature.

We thank the Reviewer and appreciate their support for the clarity of our writing.

I find that this study contributes to the quality of outbreak analysis, and I suggest the acceptance of the manuscript.

We thank the Reviewer and appreciate their support of our manuscript!

Reviewer 2 Report

Considering the difficult context and the limited quality and completeness  of the data available, the authors have done a remarkable and extensive analysis of the Cholera outbreak in Yemen. The epidemiological methods and data treatment appear adequate to the available data sets, as is their representations in tables and figures; the paper is somewhat too long and  maybe some figures may be put in the supplementary materials section. The discussion and acknowledgement of the limitations of the study are correct

Author Response

Considering the difficult context and the limited quality and completeness  of the data available, the authors have done a remarkable and extensive analysis of the Cholera outbreak in Yemen. The epidemiological methods and data treatment appear adequate to the available data sets, as is their representations in tables and figures; the paper is somewhat too long and maybe some figures may be put in the supplementary materials section. The discussion and acknowledgement of the limitations of the study are correct.

We thank the Reviewer and appreciate their support of our manuscript. While the paper is long and figures could be pushed in Supplementary Materials, Figure 1 instructs the reader how to interpret time series and derivative curves, Figure 2-3 provide comparisons of outbreak signatures across governorates, and Figures 4-9 provide signatures for each outbreak cluster. We choose to keep each of the outbreak cluster figures so that the reader can identify the unique properties of each cluster and compare properties between clusters.

Reviewer 3 Report

The graphical outbreak signature of cholera is intuitively easy to understand. It seems that the details of the outbreak can be represented in this way, making it more readily analyzed. This method may be useful for analyzing outbreaks of various infectious diseases.

As shown in Figure 1, the heat map of the smooth outbreak curve, the outbreak rate, and its derivatives is visual and very easy to understand. From the smooth curve of outbreaks in the actual case, the emergence and receding of the epidemic can also be seen at a glance. This is excellent in that it makes it possible to easily identify the waves of the epidemic.

However, in the heat map of derivative in the actual cases, for example, the color of the heat map of the 30-40 week upward curve in Figure 5 may not reflect the actual derivative value. The same thing is observed in the other figures. If the color of the heat map of derivative is emphasized, the appearance of peaks may become clearer.

he horizontal axis of each figure is the number of weeks since the start of this study, but in the text the actual date is described. It is difficult to understand when the number of weeks corresponds to. It would be easier to understand if the number of weeks were added in parentheses in the text or the date was written in the figure.

From the graph in Fig. 3, we can see the trend of outbreaks in each region, and it is easy to understand that we can classify clusters based on the trends of similar outbreaks. The propagation of the epidemic may be inferred from the location of the peak or the time of appearance of the colors in the heat map.

The number of cases per 100,000 population is used as an index, but what would be the trend in the incidence rate against population density? Have you considered this?

Author Response

The graphical outbreak signature of cholera is intuitively easy to understand. It seems that the details of the outbreak can be represented in this way, making it more readily analyzed. This method may be useful for analyzing outbreaks of various infectious diseases.

We thank the Reviewer and agree about the application of Kolmogorov-Zurbenko filters for modeling other infectious disease outbreaks. In fact, we have just submitted a new manuscript to IJERPH demonstrating the application of these filters for modeling SARS-CoV-2 test, case, and death outbreak signatures in Middlesex County, Massachusetts, USA.

As shown in Figure 1, the heat map of the smooth outbreak curve, the outbreak rate, and its derivatives is visual and very easy to understand. From the smooth curve of outbreaks in the actual case, the emergence and receding of the epidemic can also be seen at a glance. This is excellent in that it makes it possible to easily identify the waves of the epidemic.

We thank the Reviewer. This design for Figure 1 and subsequent figures thereafter was for exactly this purpose!

However, in the heat map of derivative in the actual cases, for example, the color of the heat map of the 30-40 week upward curve in Figure 5 may not reflect the actual derivative value. The same thing is observed in the other figures. If the color of the heat map of derivative is emphasized, the appearance of peaks may become clearer.

We can confirm that Figure 5 and other figures of similar design present the actual derivative values. The reason that colors appear not to match these values stems from our natural logarithm scale. For this region especially, outbreaks were characterized by enormous spikes in rates, requiring a log-transformed scale to account for the variability of rates over time and by governorate. As a result, we use the background heatmap colors to depict these changes.

The horizontal axis of each figure is the number of weeks since the start of this study, but in the text the actual date is described. It is difficult to understand when the number of weeks corresponds to. It would be easier to understand if the number of weeks were added in parentheses in the text or the date was written in the figure. From the graph in Fig. 3, we can see the trend of outbreaks in each region, and it is easy to understand that we can classify clusters based on the trends of similar outbreaks. The propagation of the epidemic may be inferred from the location of the peak or the time of appearance of the colors in the heat map.

We have modified the horizontal axis labels for Figures 2-9 to include both Week of Study (from 1-172, since Week 36 of 2016) and Calendar Date (formatted DD-Month-YY).

The number of cases per 100,000 population is used as an index, but what would be the trend in the incidence rate against population density? Have you considered this?

Thank you for your question. We did consider the examination of rates by population density. However, we decided not to pursue this consideration as: i) the large geographic area of governorates would not provide as much value as more granular spatial units, especially given many governorates included both urban city and rural mountainous areas’; and ii) the internal displacement of Yemeni citizens due to the ongoing civil war created uncertain estimates of population size and density. We make a note of this in our discussion section:

“Additionally, we were unable to adjust for possible population migration from Yemen or internal displacement within Yemen, as this data was not reported consistently at the monthly or weekly levels [36]. While a substantial proportion (~15%) of Yemen’s population has been internally displaced throughout the Yemeni Civil War, neither ACLED nor any other dataset we explored provided data on migration or internal displacement with sufficient temporal resolution, spatial granularity, or consistent reporting to use in adjusting population estimates. Furthermore, data lacked sufficient spatial resolution to consider examining rates by population density, as most governorates included both urban cities and rural mountainous areas. Because we used a prorated weekly population estimate to account for possible population growth over time in the absence of more refined population data, we likely underestimated the rate of infection.”
